# Sex Differences in the Survival of Patients with Neuroendocrine Neoplasms: A Comparative Study of Two National Databases

**DOI:** 10.3390/cancers16132376

**Published:** 2024-06-28

**Authors:** Mohamed Mortagy, Marie Line El Asmar, Kandiah Chandrakumaran, John Ramage

**Affiliations:** 1Hampshire Hospitals NHS Foundation Trust, Winchester SO22 5DG, UK; 2Internal Medicine Department, St. George University School of Medicine, West Indies, Grenada; 3Gastroenterology Department, Hampshire Hospitals NHS Foundation Trust, Basingstoke RG24 9NA, UK; marieline.elasmar@hhft.nhs.uk (M.L.E.A.);; 4Peritoneal Malignancy Institute, Hampshire Hospitals NHS Foundation Trust, Basingstoke RG24 9NA, UK; kandiah.chandrakumaran@hhft.nhs.uk; 5Faculty of Health and Wellbeing, University of Winchester, Winchester SO22 4NR, UK; 6Kings Health Partners Neuroendocrine Centre, London SE1 9RT, UK

**Keywords:** neuroendocrine tumour, neuroendocrine carcinoma, epidemiology, sex differences, cancer survival

## Abstract

**Simple Summary:**

Neuroendocrine neoplasms (NENs) are occurring more frequently worldwide. Data from the UK cancer database (National Cancer Registration and Analysis Service (NCRAS)) showed that female patients have better survival with neuroendocrine neoplasms. This study used the U.S. cancer database (Surveillance, Epidemiology, and End Results Program (SEER)) to validate and compare these findings. Sixty-months survival for NENs were calculated for both male and female patients from NCRAS and SEER. The findings from NCRAS were confirmed by the findings from SEER that females survive more than males with NENs, mainly with lung and stomach NENs. The reason behind this is unclear and remains unexplained.

**Abstract:**

Background: Neuroendocrine neoplasms (NENs) are increasing in incidence globally. Previous analysis of the UK cancer database (National Cancer Registration and Analysis Service (NCRAS)) showed a notable female survival advantage in most tumour sites. This study aims to compare NCRAS to the Surveillance, Epidemiology, and End Results Program (SEER) to validate these results using the same statistical methods. Methods: A total of 14,834 and 108,399 patients with NENs were extracted from NCRAS and SEER, respectively. Sixty-months survival for both males and females for each anatomical site of NENs were calculated using restricted mean survival time (RMST) and Kaplan–Meier Survival estimates. The sixty-month RMST female survival advantage (FSA) was calculated. Results: FSA was similar in NCRAS and SEER. The highest FSA occurred in lung and stomach NENs. Conclusions: The data from SEER confirm the findings published by NCRAS. Female survival advantage remains unexplained.

## 1. Introduction

Neuroendocrine neoplasms (NENs) consist of well-differentiated Neuroendocrine tumours (NETs) and poorly differentiated Neuroendocrine carcinomas (NECs) [1]. The incidence of NENs is increasing worldwide, with a global incidence rate ranging from 3.16 to 10 per 100,000 [2,3,4,5,6,7]. While no significant sex difference was seen in incidence, there was a notable female survival advantage in most tumour sites when analyzing the UK cancer database (National Cancer Registration and Analysis Service (NCRAS)) [8]. The reasons behind this survival advantage are unclear. Probably, hormonal, genetic, and behavioural factors may play a part. NCRAS is England’s population-based cancer registry that collects data on all patients who are diagnosed with primary tumours in England [9]. The Surveillance, Epidemiology, and End Results program (SEER) is the United States’ population-based cancer registry that collects data on cancer patients in certain states within the United States [10]. The NCRAS data only exist for England, with a population of 65 million. It seemed important to compare these findings with a larger database such as SEER, which is a population database for about 48% of the population of the USA, and for which data are posted online every year [11]. Comparing NCRAS to SEER databases could validate the previously published results from NCRAS showing a female survival advantage for NENs [8].

Most studies addressing sex survival in cancer use traditional statistical methods such as Kaplan–Meier statistics and Cox regression [7,12,13,14,15]. However, restricted mean survival time (RMST) is an alternative robust statistical method that showed better advantages when compared to the traditional survival analysis methods for many reasons [16,17,18]. RMST represents the mean expected survival time to a certain cutoff point which has better clinical interpretability. It does not rely on proportional hazards assumption, unlike Cox regression. It allows the direct comparison of mean survival time between groups. However, Cox regression is better for adjusting for multiple confounding factors but is only reliable when the proportional hazards assumptions test is met. Thus, both methods could be used in a complementary fashion for a better understanding of cancer survival. This study aims to validate the female survival advantage found in the NCRAS database using a larger database such as SEER using the same statistical method (RMST) across all the main primary NENs sites.

## 2. Materials and Methods

### 2.1. Data Sources (NCRAS and SEER)

This is a comparative population-based study that used prospectively collected data of 14,834 patients from NCRAS and 108,399 patients from SEER databases.

### 2.2. Data Extraction and Cleaning

#### 2.2.1. NCRAS

A total of 14,834 patients with NENs diagnosed between 2012 and 2018 were extracted from NCRAS. Patients with NENs were extracted using WHO ICD-O-3 morphology codes 8013 (excluding lung), 8041–8045 (excluding lung), 8150–8158, 8240–8247, 8249, and 9091 to be consistent with previously published analyses on NCRAS NEN data [2,8].

The sites included were the appendix, cecum, colon, lung, pancreas, rectum, small intestine, and stomach. Large-cell neuroendocrine tumours and small-cell carcinomas of the lung were excluded to avoid skewing the results due to their association with smoking [19,20]. Goblet cell adenocarcinomas (ICD-O-3 code 8243) were excluded as they were recently re-classified as adenocarcinoma [21]. Mixed neuroendocrine non-neuroendocrine neoplasms (MiNEN) and Merkel cell tumours were excluded. The tumour morphologies were classified as NET and NEC. NET included well-differentiated neoplasms and NEC included all carcinomas and poorly differentiated neoplasms [22]. Included variables were age, sex, stage, and morphology. NCRAS stages were reconfigured by consolidating stages I, II, III, and IV as follows: localized (stage I, II), regional (stage III), and distant (Stage IV). This aligned with the SEER database staging for direct comparison.

Original NCRAS staging was stage I (34%), stage II (13.5%), stage III (18%), and stage IV (34.5%). Staging was changed by combining stages I and II to be localized (47.5%), stage III (18%) to be regional, and stage IV to be distant (34.5%).

#### 2.2.2. SEER

A total of 289,232 patients with NEN were extracted from three databases of the surveillance, epidemiology, and end results (SEER) database by using SEER ∗Stat software version 8.4.3 [23]. Patients with NENs were extracted using WHO ICD-O-3 morphology codes 8013, 8041–8045, 8150–8158, 8240–8247 (excluding goblet cell adenocarcinoma ICD-O-3 code 8243), 8249, and 9091. WHO ICD-10 anatomy codes used for extraction were C16.0–C16.9, C17.0–C17.9, C18.0, C18.1, C18.2–C18.9, C20.9, C25.0–C25.9, and C34.0–C34.9.

A total of 31,320 patients diagnosed from 1975 to 1991 were extracted from the SEER database (Incidence—SEER Research Data, 8 Regs, Nov 2022 submission [1975–2020]) [24], and 27,301 patients diagnosed from 1992 to 1999 were extracted from the SEER database (Incidence—SEER Research Data, 12 Regs, Nov 2022 submission [1992–2020]) [25]. A total of 230,611 patients diagnosed from 2000 to 2020 were extracted from the SEER database (Incidence—SEER Research Data, 17 Regs, Nov 2022 submission [2000–2020]) [26]. Included variables were age, sex, stage, and morphology.

The extracted patients from these three databases were then combined into one cohort. This approach was used to increase the sample size from all eligible SEER databases while avoiding overlap or duplication of patients from the three SEER databases. A total of 3726 patients were excluded for not having a positive histology or cytology diagnosis. A total of 169,288 patients were excluded for having large cell neuroendocrine tumours (ICD-O3 code 8013) or small cell carcinomas of the lung (ICD-O3 codes 8041–8045). A total of 1306 patients were excluded for having mixed neuroendocrine non-neuroendocrine neoplasms (MiNEN) and Merkel cell tumours. A total of 503 patients with unknown survival duration were excluded. A total of 6010 patients who survived 0 months were excluded. The final analysis included 108,399 patients.

The morphology variable was created from the SEER AYA site recode 2020 Revision. The stage was created from the three SEER databases using two variables: SEER historic stage A (1973–2015) and combined summary stage (2004+). A flowchart detailing the stages of data extraction is shown in Appendix A.

### 2.3. Data Analysis (NCRAS and SEER)

Descriptive data were presented as frequencies and percentages for categorical variables and as the mean and standard deviation for numerical variables. Patients with missing data (0.28% of the NCRAS cohort and 8.4% of the SEER cohort) were excluded from all analyses. A *p*-value of <0.05 was considered statistically significant. Statistical analyses were performed using RStudio version 2023.12.1, Build 402, “Ocean Storm” release [27].

For each anatomical site, the predicted 60-month survival percentage for both males and females was calculated using Kaplan–Meier survival estimates. Sixty months of survival for both males and females were also calculated for each anatomical site using restricted mean survival time (RMST). Age-adjusted 60-month RMST female survival advantage was also calculated for each anatomical site. Age-adjusted hazard ratios (aHR) for the overall survival of both males and females were calculated using Cox regression for each anatomical site. The age-adjusted female-to-male 60-month ratio of the Restricted Mean Time Lost (RMTL) was calculated for each anatomical site.

Age-adjusted 60-month RMST female survival advantage (FSA) was estimated for each subgroup and for each anatomical site. The subgroups included stratified age (age ≤50 vs. age >50), morphology (NET vs. NEC), and stage (localized vs. regional vs. distant). Kaplan–Meier statistics and plots for predicted overall survival were estimated for each of the NCRAS and SEER cohorts for each site.

## 3. Results

### 3.1. Baseline Characteristics

#### 3.1.1. NCRAS

A total of 14,834 patients were included in NCRAS. The demographic characteristics of the cohort are shown in Table 1. The mean age of the cohort was 61.7 with a standard deviation of 16.2. Most patients were in the >50 age group (78.6%). The cohort had slightly more females than males (51.5 vs. 48.5%, respectively). Most patients had a localized stage (47.5%) followed by a distant stage (34.5%). Most patients had NET morphology (74.7%). The most prevalent site was the lung (31.4%), followed by the small intestine (21.6%). The least prevalent site was colon (3.4%).

#### 3.1.2. SEER

A total of 108,399 patients were included from SEER. The demographic characteristics of the cohort are shown in Table 2. The mean age of the cohort was 60.4 with a standard deviation of 14.7. Most patients were in the >50 age group (77.3%). The cohort had more females than males (53.4% vs. 46.6%, respectively). Most patients had a localized stage (50.5%) followed by a distant stage (22.6%). Most patients had NET morphology (72.4%). The most prevalent site was the lung (26.4%), followed by the small intestine (21.1%). The least prevalent site was cecum (2.6%).

### 3.2. Sex Differences in Survival

Sex differences in survival of patients with NEN from NCRAS and SEER are shown in Table 3. The table outlines 60-month Kaplan–Meier (KM) survival percentages, age-adjusted Cox regression with adjusted hazard ratios (aHR), 60-month RMST survival time in months, and age-adjusted 60-month RMST female survival advantage (FSA) over males. The age-adjusted female-to-male 60-month RMTL ratio is shown in Appendix A. The subgroup analyses of age-adjusted FSA for each site by age (≤50 vs. >50 years), morphology (NET vs. NEC), and stage (localized vs. regional vs. distant) for both NCRAS and SEER are shown in Table 4.

#### 3.2.1. All Patients

##### NCRAS

Females had higher RMST mean survival when compared to males (46.56 months (CI: 46.05–47.06) vs. 40.88 months (CI: 40.30–41.45). The 60-month FSA was 5.08 months (CI: 4.30–5.86).

Subgroup analyses showed statistically significant FSA for both the >50 and ≤50 years subgroups (5.59 vs. 3.18 months, respectively). FSA was higher in the >50 years subgroup compared to the ≤50 years subgroup. There was a statistically significant FSA for both NET and NEC morphologies (2.45 vs. 4.91 months, respectively). There was a statistically significant FSA for all stage subgroups (localized: 2.15 vs. regional: 2.28 vs. distant 3.15 months).

##### SEER

Females had higher RMST compared to males (48.82 months (CI: 48.65–48.99) vs. 45.46 months (CI: 45.26–45.66), respectively). The 60-month FSA was 3.42 months (CI: 3.17–3.67).

Subgroup analyses showed a statistically significant FSA for both the >50 and ≤50 years subgroups (3.90 vs. 2.16 months, respectively). FSA was higher in the>50 years subgroup compared to the ≤50 years subgroup. There was a statistically significant FSA for both NET and NEC morphologies (1.55 vs. 4.06 months, respectively). There was a statistically significant FSA for all stage subgroups (localized: 1.47 vs. regional: 1.70 vs. distant 4.23 months).

#### 3.2.2. Appendix

##### NCRAS

Females had similar RMST compared to males (57.25 months (CI: 56.65–57.85) vs. 55.70 months (CI: 54.76–65.65), respectively). The 60-month FSA was 1.04 months (CI: −0.07–2.15) and was not statistically significant.

Subgroup analyses showed a statistically significant FSA only for the>50 years subgroup (2.91 months), while the ≤50 years subgroup had statistically insignificant FSA (−0.01 months). There was a statistically insignificant FSA for both NET and NEC morphologies (0.91 vs. −1.8 months, respectively). There was a statistically insignificant FSA for all stage subgroups (localized: 0.99 vs. regional: 0.23 vs. distant 6.42 months).

##### SEER

Females had higher RMST compared to males (56.97 months (CI: 56.61–57.34) vs. 55.76 months (CI: 55.22–56.29), respectively). The 60-month FSA was 1.23 months (CI: 1.19–1.27).

Subgroup analyses showed a statistically significant FSA only for the>50 years subgroup (1.96 months), while the ≤50 years subgroup had a statistically insignificant FSA (0.05 months). There was a statistically significant FSA only for the NEC subgroup (3.76 months), while the NET subgroup had a statistically insignificant FSA (0.63 months). FSA was only statistically significant for the localized subgroup (1.003 months), while regional and distant subgroups had statistically insignificant FSA (0.92 and 1.67 months, respectively).

#### 3.2.3. Cecum

##### NCRAS

Females had similar RMST compared to males (39.64 months (CI: 36.77–42.50%) vs. 37.32 months (CI: 33.95–40.69%), respectively). The 60-month FSA was 2.76 months (CI: −1.62–7.14) and was statistically insignificant.

Subgroup analyses showed a statistically insignificant FSA for both the >50 and ≤50 years subgroups (2.49 vs. 12.86 months, respectively). There was a statistically insignificant FSA for both NET and NEC morphologies (1.02 vs. 0.35 months, respectively). There was a statistically insignificant FSA for all stage subgroups (localized: −2.74 vs. regional: 4.33 vs. distant 3.13 months).

##### SEER

Females had similar RMST compared to males (42.56 months (CI: 41.37–43.74) vs. 41.08 months (CI: 39.74–42.42), respectively). The 60-month FSA was 1.46 months (CI: 1.41–1.51).

Subgroup analyses showed a statistically insignificant FSA for both the >50 and ≤50 years subgroups (2.18 vs. 2.41 months, respectively). There was a statistically significant FSA only for the NET subgroup (2.07 months), while the NEC subgroup had a statistically insignificant FSA (1.05 months). There was a statistically significant FSA only for the distant subgroup (4.08 months), while localized and regional stages had statistically insignificant FSA (0.83 and 1.24 months, respectively).

#### 3.2.4. Colon

##### NCRAS

Females had similar RMST compared to males (24 months (CI: 20.58–27.41) vs. 26.67 months (CI: 23.74–29.60), respectively). The 60-month FSA was −2.33 months (CI: −6.81–2.15) and was statistically insignificant.

Subgroup analyses showed a statistically insignificant FSA for both the >50 and ≤50 years subgroups (−3.15 vs. 7.49 months, respectively). There was a statistically insignificant FSA for both NET and NEC morphologies (−6.41 vs. −0.22 months, respectively). There was a statistically insignificant FSA for all stage subgroups (localized: −4.61 vs. regional: −1.35 vs. distant −1.73 months).

##### SEER

Females had similar RMST compared to males (40.09 months (CI: 38.96–41.21) vs. 41.07 months (CI: 40–42.14), respectively). The 60-month FSA was −0.98 months (CI: −1.04–−0.92).

Subgroup analyses showed a statistically insignificant FSA for both the >50 and ≤50 years subgroups (0.46 vs. 2.11 months, respectively). There was a statistically significant FSA only for NEC morphology (2.17 months), while the NET subgroup had a statistically insignificant FSA (−0.036 months). There was a statistically insignificant FSA for all stage subgroups (localized: −0.014 vs. regional: −1.004 vs. distant 1.55 months).

#### 3.2.5. Lung

##### NCRAS

Females had higher RMST compared to males (45.08 months (CI: 44.22–45.94) vs. 35.60 months (CI: 434.40–36.80), respectively). The 60-month FSA was 9.83 months (CI: 8.38–11.29).

Subgroup analyses showed a statistically significant FSA only for the >50 years subgroup (10.83 months), while the ≤50 years subgroup showed a statistically insignificant FSA (1.92 months). There was a statistically significant FSA for both NET and NEC morphologies (4.56 vs. 6.63 months, respectively). The NET subgroup was similar to the NEC subgroup in terms of FSA. There was a statistically significant FSA for all stage subgroups (localized: 2.64 vs. regional: 7.01 vs. distant 5.39 months) and they were all overlapping and thus similar in terms of FSA.

##### SEER

Females had higher RMST compared to males (43.40 months (CI: 43.05–43.75) vs. 33.30 months (CI: 32.82–33.78), respectively). The 60-month FSA was 10.11 months (CI: 9.95–10.26).

Subgroup analyses showed a statistically significant FSA for both the >50 and ≤50 years subgroups (11.94 vs. 4.46 months, respectively). FSA was statistically higher in the>50 years subgroup compared to the ≤50 years subgroup. There was a statistically significant FSA for both NET and NEC morphologies (3.001 vs. 6.83 months, respectively). FSA was statistically higher in the NEC subgroup compared to the NET subgroup. There was a statistically significant FSA for all stage subgroups (localized: 3.72 vs. regional: 7.52 vs. distant 8.74 months). Both regional and distant subgroups overlapped, but both had higher FSA compared to localized subgroups.

#### 3.2.6. Pancreas

##### NCRAS

Females had higher RMST compared to males (42.24 months (CI: 40.72–43.77) vs. 37.98 months (CI: 36.57–39.38), respectively). The 60-month FSA was 3.77 months (CI: 1.68–5.86).

Subgroup analyses showed a statistically significant FSA for both the >50 and ≤50 years subgroups (2.65 vs. 8.63 months, respectively). Both the >50 and ≤50 years subgroups had similar FSA. There was a statistically significant FSA for NEC, while NET morphology had a statistically insignificant FSA (4.23 vs. 1.44 months, respectively). There was statistically insignificant FSA for the localized stage only (2.02 months), while regional and distant subgroups had statistically insignificant FSA. (3.04 and. 2.36 months, respectively).

##### SEER

Females had higher RMST compared to males (43.67 months (CI: 43.09–44.24) vs. 41.57 months (CI: 41.04–42.11), respectively). The 60-month FSA was 2.11 months (CI: 2.0–2.21).

Subgroup analyses showed a statistically significant FSA for both the >50 and ≤50 years subgroups (1.59 vs. 2.19 months, respectively). Both the >50 and ≤50 years subgroups had similar FSA. There was a statistically significant FSA for both NET and NEC morphologies (1.61 vs. 1.37 months, respectively). Both the NET and NEC subgroups had similar FSA. There was a statistically significant FSA only for the localized stage (0.91 months), while the regional and distant stages had statistically insignificant FSA (1.13 and 0.71 months, respectively).

#### 3.2.7. Rectum

##### NCRAS

Females had higher RMST compared to males (44.90 months (CI: 42.57–47.22) vs. 38.87 months (CI: 36.65–41.08), respectively). The 60-month FSA was 5.61 months (CI: 2.32–8.91).

Subgroup analyses showed a statistically significant FSA only for the >50 years subgroups (6.50 months), while the ≤50 subgroup had a statistically insignificant FSA (−0.16 months). There was a statistically significant FSA only for NET morphology (2.31 months), while the NEC subgroup had a statistically insignificant FSA (1.62 months). There was a statistically significant FSA only for the localized stage subgroup (2.0 months), while regional and distant subgroups did not have significant FSA (1.15 and 2.31 months, respectively).

##### SEER

Females had better RMST compared to males (56.49 months (CI: 56.25–56.73) vs. 54.92 months (CI: 54.64–55.21), respectively). The 60-month FSA was 1.57 months (CI: 1.49–1.64).

Subgroup analyses showed a statistically significant FSA for both the >50 and ≤50 years subgroups (1.72 vs. 1.11 months, respectively). Both the >50 and ≤50 subgroups had similar FSA. There was a statistically significant FSA for both the NET and NEC morphologies (0.92 vs. 3.10 months, respectively). Both the NET and NEC subgroups had similar FSA. There was a statistically higher FSA only for the localized stage subgroup (0.83 months), while the regional and distant stages did not have statistically significant FSA (1.54 and 2.38 months, respectively).

#### 3.2.8. Small Intestine

##### NCRAS

Females had similar RMST compared to males (49.13 months (CI: 48.07–50.19) vs. 48.16 months (CI: 47.20–49.11), respectively). The 60-month FSA was 1.31 months (CI: −0.13–2.76).

Subgroup analyses showed a statistically significant FSA for the ≤50 years subgroup (3.43 months). The >50 years subgroup showed a statistically insignificant FSA (1.11 months). Both the NET and NEC morphologies had statistically insignificant FSA (1.29 vs. 2.58 months, respectively). There was a statistically insignificant FSA for all stage subgroups (localized: 1.50 vs. regional: 1.31 vs. distant 1.33 months).

##### SEER

Females had higher RMST compared to males (51.87 months (CI: 51.55–52.20) vs. 50.99 months (CI: 50.67–51.32), respectively). The 60-month FSA was 0.88 months (CI: 0.78–0.98).

Subgroup analyses showed a statistically significant FSA for the >50 years subgroups (1.09 months). FSA for the ≤50 years subgroup was statistically insignificant (0.57 months). There was a statistically significant FSA for the NET subgroup (0.91 months). FSA for the NEC subgroup was statistically insignificant (1.08 months). There was a statistically significant FSA for the localized and distant subgroups, and both subgroups overlapped (1.36 and 1.19 months, respectively). FSA was statistically insignificant for the regional stage (0.57 months).

#### 3.2.9. Stomach

##### NCRAS

Females had higher RMST compared to males (38.75 months (CI: 35.54–41.95) vs. 26.09 months (CI: 23.61–28.57), respectively). The 60-month FSA was 10.32 months (CI: 6.12–14.52).

Subgroup analyses showed a statistically significant FSA for both the >50 and ≤50 years subgroups (9.54 vs. 12.81 months, respectively), which overlapped. There was a statistically insignificant FSA for both the NET and NEC morphologies (4.59 vs. 0.14 months, respectively). There was a statistically insignificant FSA for all stage subgroups (localized: 3.47 vs. regional: 5.71 vs. distant 1.32 months).

##### SEER

Females had higher RMST compared to males (51.16 months (CI: 50.65–51.67) vs. 42.98 months (CI: 42.19–43.77, respectively). The 60-month FSA was 8.20 months (CI: 8.12–8.27).

Subgroup analyses showed a statistically significant FSA for both the >50 and ≤50 years subgroups (8.37 vs. 7.36 months, respectively), and both overlapped. There was a statistically significant FSA for both the NET and NEC morphologies (3.45 vs. 10.96 months, respectively). The NEC subgroup showed a higher FSA than the NET subgroup. There was a statistically significant FSA for localized and distant subgroups, and both subgroups overlapped (3.0 and 5.31 months, respectively). Distant subgroups had higher FSA than localized subgroups. FSA was statistically insignificant for the regional stage (2.91 months).

### 3.3. Kaplan–Meier (KM) Graphs

Figure 1 and Figure 2 show the non-adjusted KM graphs for the overall survival of NEN patients for each site for both SEER and NCRAS cohorts, respectively. The site with the highest overall survival for SEER and NCRAS cohorts was the appendix. The sites with the lowest overall survival for SEER and NCRAS cohorts were lung and colon, respectively. Figure 3 and Figure 4 show the non-adjusted KM graphs for each site subgrouped by sex for both NCRAS and SEER, respectively. NCRAS cohort showed a statistically higher female survival advantage for the appendix, lung, pancreas, rectum, and stomach sites on KM curves and associated log-rank tests. SEER cohort showed a statistically higher female survival advantage for the appendix, lung, pancreas, rectum, small intestine, and stomach sites on KM curves and associated log-rank tests.

## 4. Discussion

### 4.1. Summary of Findings

To the best of the authors’ knowledge, this is the first population-based study comparing both SEER and NCRAS databases using RMST analysis. Overall, the largest FSA seemed to occur in the lung and stomach primary sites in both cohorts. In the NCRAS cohort, the stomach, lung, rectum, and pancreas NEN sites showed statistically significant age-adjusted FSA. The stomach showed the highest FSA (10.32 months) and the pancreas showed the lowest FSA (3.77 months). Subgroup analyses showed higher FSA in the >50 years subgroup in lung and rectum sites and similar FSA in the >50 and ≤50 subgroups in pancreas and stomach FSA. NET and NEC subgroups showed similar FSA in the lung and stomach sites. However, NEC had higher FSA in the pancreas and NET had higher FSA in the rectum. All stage subgroups had similar FSA in the lung and stomach. However, the localized stage showed higher FSA in the pancreas and rectum.

In the SEER cohort, the appendix, cecum, lung, pancreas, rectum, small intestine, and stomach NEN sites showed statistically significant age-adjusted FSA. The lung showed the highest FSA (10.11 months), and the small intestine showed the lowest FSA (0.88 months). Subgroup analyses showed similar FSA in both the >50 and ≤50 years subgroups in all the aforementioned sites; however, the appendix and lung have higher FSA in the >50 years subgroup. NET had higher FSA in the cecum and small intestine. However, NEC had higher FSA in the appendix, stomach, and lung. NET and NEC had similar FSA in the pancreas. The localized stage had higher FSA than other stages in the pancreas and rectum. The appendix had similar FSA in all stage subgroups. Localized and distant subgroups had higher FSA compared to regional subgroups in the stomach and small intestine. However, the regional and distant sites had better FSA compared to the localized subgroup in the lung.

There was a similarity of findings in NCRAS vs. SEER in terms of FSA, with the highest differences occurring in lung and stomach NEN. Overall, this validates the original findings in NCRAS data previously published.

### 4.2. Previous Studies of Sex Differences in Neuroendocrine Tumours

Previous studies have demonstrated that females generally have a survival advantage over males for NENs [28,29,30,31]. This study reflects the same findings using data from both the NCRAS and SEER cohorts. Specifically, evidence suggests that females have a survival advantage over males for lung, rectal, pancreatic, and gastric NENs in England. However, females lack a survival advantage over males for appendiceal, caecal, colonic, and small intestinal NENs [8]. Additionally, females were previously shown to have a survival advantage in lung [32,33], appendiceal [34,35], and gastro-pancreatic NEN [36], which was also shown by this study. There is currently no clear underlying explanation for such female survival advantage [37].

### 4.3. Possible Explanations for Female Survival Advantage in NENs

#### 4.3.1. Hormonal Receptors Positivity

A study showed that NENs show focal to diffuse estrogen and progesterone expression using immunohistochemical analysis [38]. All primary NEN sites showed expression of hormonal receptors, except the colon and rectum [39]. Specifically, pancreatic NETs with better outcomes had higher estrogen receptor-β expression [40], while the loss of progesterone receptor expression was associated with higher grade, larger tumours, and decreased patient survival time [41]. Estrogen receptor expression was found to have higher expression in non-pancreatic NETs in females compared to males [42]. Additionally, a study showed that higher body mass index (BMI) is associated with better overall survival in patients with NENs [43]. This could be related to higher estrogen levels among patients with higher BMI [44].

Furthermore, a single-arm study showed Tamoxifen use in estrogen receptor/progesterone receptor-positive patients to be safe, but there was no clear effect and the trial was terminated [45]. Another study showed that inhibiting estrogen receptor-α with Fluvestrant and SiRNA increases the radiosensitivity of NETs [46]. It is also possible that androgens could influence survival in males, as androgen receptors are present in certain tumours such as prostate cancer [47], but this has not been investigated in NENs to the best of the authors’ knowledge.

#### 4.3.2. Behavioural Differences between Sexes

Another explanation for female survival advantage is the difference in health behaviours among the sexes. Males typically delay seeking medical help compared to females [48,49,50]. However, this does not account for the differences in survival across various NEN sites. While differences in health behaviour may contribute to the observed sex-survival difference, this factor alone is unlikely to be a significant cause.

#### 4.3.3. Preponderance of Different Tumours

Regarding lung NENs, diffuse idiopathic pulmonary neuroendocrine cell hyperplasia (DIPNECH) is more common in females and tends to have a good prognosis. This could explain the higher survival in females with lung NENs. However, the incidence of DIPNECH is low and is unlikely to affect the overall survival of lung NENs [51,52]. Regarding gastric NENs, autoimmune gastritis is more common in females [53] and is associated with type 1 gastric carcinoids [54] which have a better prognosis than other types [55]. There were in fact more localized stage gastric NENs in females in both NCRAS and SEER cohorts, which may be a factor. However, FSA was only significant in the SEER cohort, and thus it is not a consistent factor affecting the sex difference.

### 4.4. Strengths, Limitations, and Implications

This is the largest population-based study comparing sex differences in survival in patients with NENs using two large national databases with reliable follow-ups. Another strength is the use of the same robust statistical method such as RMST to compare the two groups (males vs. females) in terms of survival in both the NCRAS and SEER cohorts for the first time. This adds to the validation of previously published results regarding survival advantage in female patients with NENs. The limitations include the absence of detailed tumour pathology characteristics (such as ki-67) and hormonal receptor status in both databases. Moreover, there were no data on the behaviour of patients. Therapy variables (e.g., somatostatin analogues therapy and calcineurin inhibitor therapy) were missing and/or inaccurate in both databases. Therapy could have an effect on the quality of life and prognosis [56]. These databases did not have any data on the migration of patients. Some variables in SEER have limitations due to incomplete, missing data and/or imprecision [57].

## 5. Conclusions

The data from SEER largely confirm the findings published previously from NCRAS. The female survival advantage remains unexplained. The robust differences in sex survival in some NEN sites require further studies, including detailed tumour pathology. These sex differences may shed some light on the etiology and/or behaviour of these tumours, which is poorly understood at present. The investigation of estrogen and androgen receptors, healthcare behaviour, and genetics of these tumours may give some explanations in the future. Trials of new therapies, such as androgen receptor antagonists, may be justified.

## Figures and Tables

**Figure 1 cancers-16-02376-f001:**
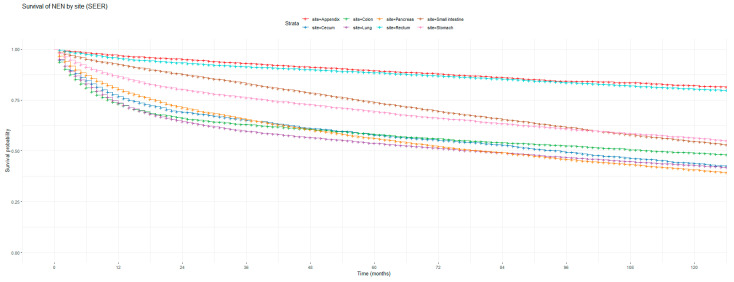
Kaplan–Meier graph for the overall survival of NEN patients in the SEER cohort (classified by site, not age adjusted).

**Figure 2 cancers-16-02376-f002:**
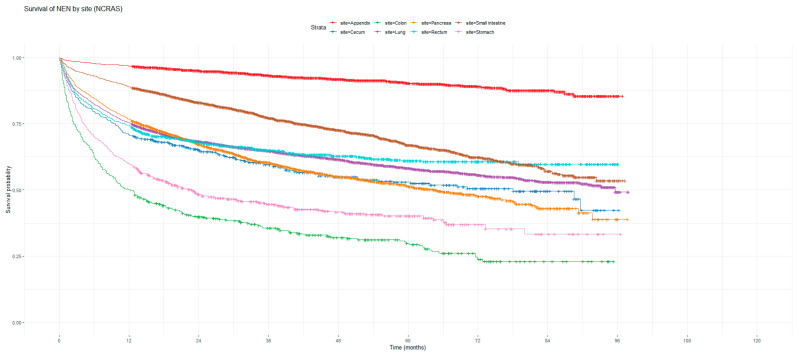
Kaplan–Meier graph for the overall survival of NEN patients in the NCRAS cohort (classified by site, not age adjusted).

**Figure 3 cancers-16-02376-f003:**
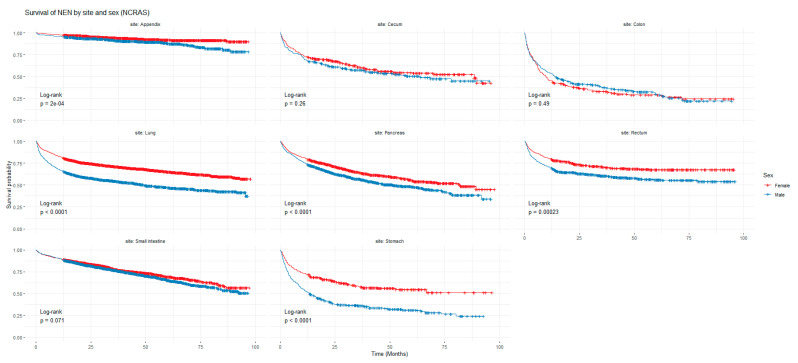
Kaplan-Meier graph for patients with NENs in the NCRAS cohort (classified by site and sex). The shown *p*-value is for the difference in survival between males and females for each anatomical site using a log-rank test (not age-adjusted).

**Figure 4 cancers-16-02376-f004:**
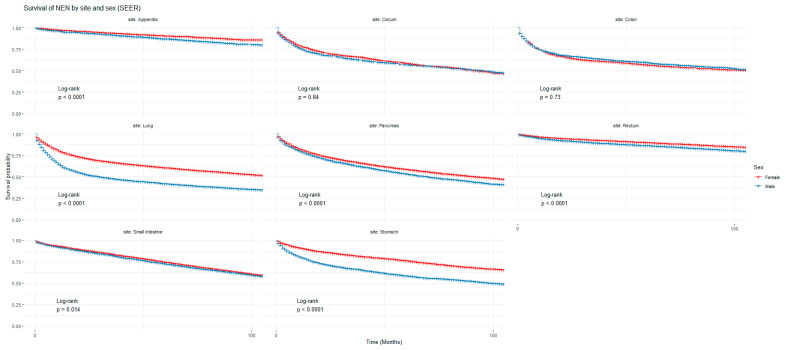
Kaplan–Meier graph for patients with NENs in the SEER cohort (classified by site and sex). The shown *p*-value is for the difference in survival between males and females for each anatomical site using a log-rank test (not age-adjusted).

**Table 1 cancers-16-02376-t001:** NCRAS cohort baseline characteristics.

	Appendix (N = 2146)	Cecum (N = 528)	Colon (N = 509)	Lung (N = 4661)	Pancreas (N = 2183)	Rectum (N = 948)	Small Intestine (N = 3201)	Stomach (N = 658)	Overall (N = 14,834)
Age									
Mean (SD)	43.9 (20.0)	66.0 (11.7)	67.5 (13.6)	64.9 (13.6)	61.7 (13.7)	60.5 (14.6)	66.4 (12.0)	67.8 (12.7)	61.7 (16.2)
Age bands									
>50	806 (37.8%)	475 (90.1%)	450 (88.6%)	4002 (86.0%)	1711 (78.5%)	733 (77.7%)	2863 (89.7%)	584 (89.0%)	11,624 (78.6%)
≤50	1327 (62.2%)	52 (9.9%)	58 (11.4%)	652 (14.0%)	468 (21.5%)	210 (22.3%)	330 (10.3%)	72 (11.0%)	3169 (21.4%)
Sex									
Male	831 (38.7%)	228 (43.2%)	298 (58.5%)	1854 (39.8%)	1230 (56.3%)	544 (57.4%)	1804 (56.4%)	407 (61.9%)	7196 (48.5%)
Female	1315 (61.3%)	300 (56.8%)	211 (41.5%)	2807 (60.2%)	953 (43.7%)	404 (42.6%)	1397 (43.6%)	251 (38.1%)	7638 (51.5%)
Stage									
Distant	60 (2.8%)	269 (50.9%)	317 (62.3%)	1402 (30.1%)	1082 (49.6%)	272 (28.7%)	1421 (44.4%)	298 (45.3%)	5121 (34.5%)
Localized	1857 (86.5%)	42 (8.0%)	79 (15.5%)	2843 (61.0%)	882 (40.4%)	587 (61.9%)	484 (15.1%)	270 (41.0%)	7044 (47.5%)
Regional	229 (10.7%)	217 (41.1%)	113 (22.2%)	416 (8.9%)	219 (10.0%)	89 (9.4%)	1296 (40.5%)	90 (13.7%)	2669 (18.0%)
Morphology									
NEC	78 (3.6%)	154 (29.2%)	316 (62.1%)	1638 (35.1%)	640 (29.3%)	333 (35.1%)	271 (8.5%)	324 (49.2%)	3754 (25.3%)
NET	2068 (96.4%)	374 (70.8%)	193 (37.9%)	3023 (64.9%)	1543 (70.7%)	615 (64.9%)	2930 (91.5%)	334 (50.8%)	11,080 (74.7%)

**Table 2 cancers-16-02376-t002:** SEER cohort baseline characteristics.

	Appendix (N = 6609)	Cecum (N = 2854)	Colon (N = 4067)	Lung (N = 28,610)	Pancreas (N = 14,340)	Rectum (N = 20,341)	Small Intestine (N = 22,904)	Stomach (N = 8674)	Overall (N = 108,399)
Age bands									
>50	2415 (36.5%)	2398 (84.0%)	3283 (80.7%)	24,077 (84.2%)	11,148 (77.7%)	14,244 (70.0%)	19,301 (84.3%)	6937 (80.0%)	83,803 (77.3%)
≤50	4194 (63.5%)	456 (16.0%)	784 (19.3%)	4533 (15.8%)	3192 (22.3%)	6097 (30.0%)	3603 (15.7%)	1737 (20.0%)	24,596 (22.7%)
Age									
Mean (SD)	42.9 (19.2)	63.4 (13.1)	62.1 (13.6)	63.7 (13.9)	60.7 (13.9)	55.9 (11.8)	63.7 (12.9)	62.5 (13.6)	60.4 (14.7)
Sex									
Female	4055 (61.4%)	1562 (54.7%)	1942 (47.8%)	17,565 (61.4%)	6413 (44.7%)	10,122 (49.8%)	10,981 (47.9%)	5234 (60.3%)	57,874 (53.4%)
Male	2554 (38.6%)	1292 (45.3%)	2125 (52.2%)	11,045 (38.6%)	7927 (55.3%)	10,219 (50.2%)	11,923 (52.1%)	3440 (39.7%)	50,525 (46.6%)
Stage									
Distant	259 (3.9%)	1132 (39.7%)	1129 (27.8%)	9011 (31.5%)	5757 (40.1%)	854 (4.2%)	5364 (23.4%)	981 (11.3%)	24,487 (22.6%)
Localized	4786 (72.4%)	378 (13.2%)	1721 (42.3%)	12,616 (44.1%)	5362 (37.4%)	16,338 (80.3%)	7886 (34.4%)	5668 (65.3%)	54,755 (50.5%)
Regional	1444 (21.8%)	1254 (43.9%)	732 (18.0%)	5218 (18.2%)	2739 (19.1%)	492 (2.4%)	8325 (36.3%)	542 (6.2%)	20,746 (19.1%)
Missing	120 (1.8%)	90 (3.2%)	485 (11.9%)	1765 (6.2%)	482 (3.4%)	2657 (13.1%)	1329 (5.8%)	1483 (17.1%)	8411 (7.8%)
Morphology									
NEC	714 (10.8%)	948 (33.2%)	1569 (38.6%)	11,780 (41.2%)	6494 (45.3%)	2252 (11.1%)	3554 (15.5%)	1910 (22.0%)	29,221 (27.0%)
NET	5891 (89.1%)	1906 (66.8%)	2489 (61.2%)	16,824 (58.8%)	7335 (51.2%)	18,033 (88.7%)	19,260 (84.1%)	6709 (77.3%)	78,447 (72.4%)
Missing	4 (0.1%)	0 (0%)	9 (0.2%)	6 (0.0%)	511 (3.6%)	56 (0.3%)	90 (0.4%)	55 (0.6%)	731 (0.7%)

**Table 3 cancers-16-02376-t003:** 60-month Kaplan–Meier survival percentage, and age-adjusted hazard ratio, 60-month RMST survival (with age-adjusted RMST female survival advantage) for patients with NEN in NCRAS and SEER cohorts (classified by sex and site) ^1^.

Site	Sex	60-Month KM Survival (Percentage)		60-Month RMST Survival (Months)		Age-Adjusted Cox Regression (aHR)		Age-Adjusted 60-Month RMST Female Survival Advantage (FSA)	
		%	CI	Months	CI	aHR	*p*-Value	Months	CI
ALL (UK)	M	56	54.60–57.30	40.88	40.30–41.45	1.50	<0.001 *	5.08	4.30–5.86
	F	67.9	66.70–69.10	46.56	46.05–47.06	0.66	<0.001 *		
ALL (USA)	M	64.80	64.30–65.20	45.46	45.26–45.66	1.32	<0.001 *	3.42	3.17–3.67
	F	71.70	71.30–72.10	48.82	48.65–48.99	0.75	<0.001 *		
Appendix (UK)	M	88.02	85.40–90.72	55.70	54.76–56.65	1.07	<0.001 *	1.04	−0.07–2.15
	F	92.0	90.3–93.8	57.25	56.65–57.85	0.62	0.001 *		
Appendix (USA)	M	87.20	85.60–88.80	55.76	55.22–56.29	1.39	<0.001 *	1.23	1.19–1.27
	F	90.80	89.70–91.90	56.97	56.61–57.34	0.71	<0.001 *		
Cecum (UK)	M	50.6	43.9–58.4	37.32	33.95–40.69	1.17	0.231	2.76	−1.62–7.14
	F	54.77	48.80–61.47	39.64	36.77–42.50	0.85	0.231		
Cecum (US)	M	56.83	54.09–59.72	41.08	39.74–42.42	1.07	0.147	1.46	1.41–1.51
	F	58.40	55.90–61.00	42.56	41.37–43.74	0.93	0.147		
Colon (UK)	M	30.22	24.8–36.81	26.67	23.74–29.60	0.94	0.599	−2.33	−6.81–2.15
	F	29.24	23.35–36.63	24.00	20.58–27.41	1.05	0.599		
Colon (US)	M	59.40	57.30–61.60	41.07	40.00–42.14	1.08	0.06	−0.98	−1.04–0.92
	F	56.81	54.57–59.15	40.09	38.96–41.21	0.92	0.06		
Lung (UK)	M	47.39	44.86–50.05	35.60	34.40–36.80	1.89	<0.001 *	9.83	8.38–11.29
	F	65.33	63.36–67.37	45.08	44.22–45.94	0.52	<0.001 *		
Lung (US)	M	42.30	41.40–43.30	33.30	32.82–33.78	1.80	<0.001 *	10.11	9.95–10.26
	F	60.90	60.20–61.70	43.40	43.05–43.75	0.55	<0.001 *		
Pancreas (UK)	M	48.26	45.00–51.76	37.98	36.57–39.38	1.25	<0.001 *	3.77	1.68–5.86
	F	55.77	52.01–59.80	42.24	40.72–43.77	0.79	<0.001 *		
Pancreas (US)	M	53.70	52.50–55.00	41.57	41.04–42.11	1.12	<0.001 *	2.11	2.00–2.21
	F	59.00	57.70–60.40	43.67	43.09–44.24	0.88	<0.001 *		
Rectum (UK)	M	56.20	51.66–61.15	38.87	36.65–41.08	1.45	0.001 *	5.61	2.32–8.91
	F	67.56	62.64–72.86	44.90	42.57–47.22	0.68	0.001 *		
Rectum (US)	M	86.60	85.90–87.30	54.92	54.64–55.21	1.36	<0.001 *	1.57	1.49–1.64
	F	90.20	89.60–90.80	56.49	56.25–56.73	0.73	<0.001 *		
Small intestine (UK)	M	65.34	62.54–68.28	48.16	47.20–49.11	1.19	0.009 *	1.31	−0.13–2.76
	F	69.10	66.07–72.28	49.13	48.07–50.19	0.83	0.009 *		
Small intestine (US)	M	72.80	71.90–73.70	50.99	50.67–51.32	1.11	<0.001 *	0.88	0.78–0.98
	F	74.90	74.00–75.80	51.87	51.55–52.20	0.89	<0.001 *		
Stomach (UK)	M	31.54	26.95–36.91	26.09	23.61–28.57	1.84	<0.001 *	10.32	6.12–14.52
	F	54.80	48.20–62.30	38.75	35.54–41.95	0.54	<0.001 *		
Stomach (US)	M	58.60	56.90–60.40	42.98	42.19–43.77	1.71	<0.001 *	8.20	8.12–8.27
	F	76.30	75.00–77.60	51.16	50.65–51.67	0.58	<0.001 *		

^1^ Patients with missing data were excluded from statistical analyses. * statistically significant.

**Table 4 cancers-16-02376-t004:** 60-month RMST age-adjusted female survival advantage for patients with NENs in NCRAS and SEER cohorts (sub-grouped by age, morphology, and stage) ^1^.

Subgroup	Age > 50	Age < 50	NET	NEC	Localized Stage	Regional Stage	Distant Stage
All (UK)	5.59 (4.70–6.48)	3.18 (2.03–4.33)	2.45 (1.76–3.15)	4.91 (3.45–6.37)	2.15 (1.48–2.82)	2.28 (0.63–3.93)	3.15 (1.80–4.51)
All (USA)	3.90 (3.59–4.21)	2.16 (1.71–2.60)	1.55 (1.31–1.78)	4.06 (3.48–4.63)	1.47 (1.23–1.71)	1.70 (1.16–2.24)	4.23 (3.62–4.83)
Appendix (UK)	2.91 (0.40–5.42)	−0.01 (−0.61–0.58)	0.91 (−0.16–1.98)	−1.8 (−11.9–8.33)	0.99 (−0.08–2.07)	0.23 (−2.83–3.31)	6.42 (−6.03–18.87)
Appendix (USA)	1.96 (0.52–3.39)	0.05 (−0.36–0.47)	0.63 (−0.02–1.28)	3.76 (1.20–6.32)	1.003 (0.36–1.64)	0.92 (−0.41–2.27)	1.67 (−4.01–7.36)
Cecum (UK)	2.49 (−2.08–7.06)	12.86 (−1.45–27.17)	1.02 (−0.09–2.14)	0.35 (−6.4–7.11)	−2.74 (−14.22–8.73)	4.33 (−1.35–10.02)	3.13 (−3.10–9.37)
Cecum (USA)	2.18 (−0.29–4.07)	2.41 (−1.86–6.69)	2.07 (0.30–3.84)	1.05 (−2.20–4.30)	0.83 (−2.60–4.28)	1.24 (−0.76–3.25)	4.08 (1.26–6.91)
Colon (UK)	−3.15 (−7.87–1.56)	7.49 (−6.57–21.55)	−6.41 (−13.27–0.43)	−0.22 (−4.90–4.46)	−4.61 (−12.87–3.64)	−1.35 (−10.74–8.02)	−1.73 (−6.27–2.80)
Colon (USA)	0.46 (−1.29–2.22)	2.11 (−1.83–6.06)	−0.36 (−1.97–1.25)	2.17 (−0.26–4.60)	−0.14 (−1.28–1.006)	−1.004 (−4.46–2.45)	1.55 (−0.59–3.70)
Lung (UK)	10.83 (9.24–12.41)	1.92 (−0.60–4.46)	4.56 (3.18–5.94)	6.63 (4.49–8.76)	2.64 (1.51–3.76)	7.01 (2.45–11.57)	5.39 (3.46–7.32)
Lung (USA)	11.94 (11.28–12.59)	4.46 (3.25–5.67)	3.001 (2.41–3.58)	6.83 (6.05–7.62)	3.72 (3.11–4.33)	7.52 (6.29–8.74)	8.74 (7.92–9.56)
Pancreas (UK)	2.65 (0.29–5.01)	8.63 (4.46–12.80)	1.44 (−0.68–3.57)	4.23 (0.59–7.88)	2.02 (0.01–4.03)	3.04 (−2.97–9.05)	2.36 (−0.51–5.24)
Pancreas (USA)	1.59 (0.67–2.51)	2.19 (0.63–3.75)	1.61 (0.65–2.57)	1.37 (0.15–2.58)	0.97 (0.16–1.78)	1.13 (−0.42–2.69)	0.71 (−0.53–1.97)
Rectum (UK)	6.50 (2.75–10.26)	−0.16 (−5.24–4.91)	2.31 (0.29–4.33)	1.62 (−2.63–5.87)	2.00 (0.29–3.70)	1.15 (−9.52–11.83)	2.31 (−1.68–6.31)
Rectum (USA)	1.72 (1.24–2.19)	1.11 (0.37–1.86)	0.92 (0.61–1.23)	3.10 (0.98–5.22)	0.83 (0.55–1.11)	1.54 (−2.71–5.79)	2.38 (−0.38–5.16)
Small intestine (UK)	1.11 (−0.43–2.66)	3.43 (0.69–6.17)	1.29 (−0.16–2.75)	2.58 (−3.25–8.41)	1.50 (−1.61–4.62)	1.31 (−0.57–3.21)	1.33 (−1.05–3.72)
Small intestine (USA)	1.09 (0.58–1.61)	0.57 (−0.25–1.39)	0.91 (0.41–1.40)	1.08 (−0.16–2.32)	1.36 (0.60–2.11)	0.57 (−0.06–1.20)	1.19 (0.14–2.25)
Stomach (UK)	9.54 (5.24–13.84)	12.81 (1.75–23.87)	4.59 (−0.12–9.31)	0.14 (−4.79–5.07)	3.47 (−1.22–8.16)	5.71 (−6.91–18.33)	1.32 (−3.39–6.04)
Stomach (USA)	8.37 (7.22–9.52)	7.36 (5.29–9.43)	3.45 (2.48–4.43)	10.96 (8.60–13.32)	3.00 (2.09–3.91)	2.91 (−1.13–6.97)	5.31 (2.54–8.07)

^1^ Patients with missing data were excluded from statistical analyses.

## Data Availability

NCRAS: DARS-NIC-656877-H3Z0P-v1.4 The data are only available upon applying through the Data Access Request Service (DARS) [https://digital.nhs.uk/services/data-access-request-service-dars (accessed on 25 February 2021)], which is administered by NHS ENGLAND. SEER: The datasets generated and analyzed in this study are anonymized public patient data and are available in the U.S. SEER database. Ethics clearance and informed patient consent were not required as the study involved secondary data analysis. The datasets are accessible for free at the following link: [https://seer.cancer.gov/ (accessed on 24 March 2024)].

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
