# Peer review of "Sex Differences in the Survival of Patients with Neuroendocrine Neoplasms: A Comparative Study of Two National Databases"

_cancers, 2024, doi:10.3390/cancers16132376_

Round 1

Reviewer 1 Report

Comments and Suggestions for Authors

I have carefully read the article by our colleagues several times, who compared the SEER and NCRAS databases on neuroendocrine tumors. The differentiation between NEC and NET with the exclusion of MiNEN is very correct, to make the group homogeneous such as the exclusion of small cell lung tumors and goblet cell adenocarcinoma. Patient numbers are considerably high, but we now know that there is a certain resurgence of these neoplasms once known as carcinoids. The researchers did not say anything about the treatment of these tumors, we know nothing about Ki 67 (a weakness admitted by themselves) but we also know nothing about the therapies carried out, somatostatin and analogues, with any complications that may affect the quantity of life (PMID: 38051513 if you believe you can mention it), on possible therapies with calcineurin inhibitor everolimus, on therapies carried out with immunotrapic drugs. Researchers have noticed that for some of these neuroendocrine neoplasms there is a higher survival rate for females than for males and therefore have hypothesized the possibility that estrogen-progestin prevalence has a significance in this pathology. In reality, numerous studies indicate how chemokines and cytokines have the their considerable importance together with the immune system in inducing this pathology. (https://doi.org/10.3390/cancers16101799 if you want, you can cite it). In conclusion, in my opinion the paper should be revised in light of these suggestions which can certainly improve it: The iconography is excellent, the bibliography can be improved, the English is excellent.

Author Response

Comment 1:
I have carefully read the article by our colleagues several times, who compared the SEER and NCRAS databases on neuroendocrine tumors. The differentiation between NEC and NET with the exclusion of MiNEN is very correct, to make the group homogeneous such as the exclusion of small cell lung tumors and goblet cell adenocarcinoma. Patient numbers are considerably high, but we now know that there is a certain resurgence of these neoplasms once known as carcinoids. 

Response 1:
We appreciate the reviewer taking the time to review our manuscript. The comments are thoughtful and helpful in improving the manuscript. We appreciate that the reviewer agrees with our methods and patient choice. We have high patient numbers as mentioned because we included 2 large databases of NEN patients in 8 different anatomical sites.

Comment 2
The researchers did not say anything about the treatment of these tumors, we know nothing about Ki 67 (a weakness admitted by themselves) but we also know nothing about the therapies carried out, somatostatin and analogues, with any complications that may affect the quantity of life (PMID: 38051513 if you believe you can mention it), on possible therapies with calcineurin inhibitor everolimus, on therapies carried out with immunotrapic drugs. 

Response 2
We indeed did not have Ki67 as a variable in our 2 cohorts which is a limitation listed and unfortunately, we could not include Ki67 as it is not available in the databases despite agreeing with Ki67 importance. We agree that therapies (e.g., somatostatin analogues, calcineurin inhibitor) are important variables. However, The UK NCRAS database lacked these variables. The U.S. SEER database had treatment variables only for surgery types and intake of chemotherapy (Yes/No/Unknown) with no details to specific classes. However, Therapy variables in SEER have some inaccuracies and missing variables. We already cited this as a limitation in our manuscript (DOI:10.1097/MLR.0000000000000073). We do not believe there is a systematic bias between treatment of males and females with NEN in the USA and UK. Thank you for suggesting PMID: 38051513 as a reference. We cited it in the limitations section as treatment and quality of life could have an effect on prognosis. 

Comment 3
Researchers have noticed that for some of these neuroendocrine neoplasms there is a higher survival rate for females than for males and therefore have hypothesized the possibility that estrogen-progestin prevalence has a significance in this pathology. In reality, numerous studies indicate how chemokines and cytokines have the their considerable importance together with the immune system in inducing this pathology. (https://doi.org/10.3390/cancers16101799 if you want, you can cite it). In conclusion, in my opinion the paper should be revised in light of these suggestions which can certainly improve it: The iconography is excellent, the bibliography can be improved, the English is excellent.

Response 3
Thank you for pointing our attention to the importance of chemokines and cytokines that could affect the immune system in NENs. We do not think this is different between males and females from what we can tell from the literature, so this would not represent a consistent bias related to sex difference. We appreciate the comments.

Reviewer 2 Report

Comments and Suggestions for Authors

very intersting research, perhaps add tables  or graphics as supplementray metrial and sinthetize results, they are very long

Author Response

Comments 1

very intersting research, perhaps add tables  or graphics as supplementray metrial and sinthetize results, they are very long

Response 1

We appreciate taking the time to review our manuscript. The comments are thoughtful and helpful in improving the manuscript. We moved Figure 1 to the supplementary section. We agree that the results and tables are long because the analysis included 2 large national databases with 8 different anatomical sites and many subgroups. Hence, we moved the following column of table 3 to the supplementary section as table 1 (supplementary material): Age-adjusted female to male 60-months RMTL ratio. We appreciate the comments.

Reviewer 3 Report

Comments and Suggestions for Authors

1. Gender and sex words should not be used interchangeably. Sex differences are not the same as gender differences. Please correct word gender differences from the 1. title of the manuscript  and  2. line 157, line 167, line 411, 440, 442 to "sex differences".

2. Figure 2 and fig 3 graphs and table formatting is not consistent. The grid format is present in the fig 2 but not fig 3. 

Author Response

Comment 1

  1. Gender and sex words should not be used interchangeably. Sex differences are not the same as gender differences. Please correct word gender differences from the 1. title of the manuscript and  2. line 157, line 167, line 411, 440, 442 to "sex differences".

Response 1

Thank you for taking the time to review our manuscript. We agree that gender and sex should not be used interchangeably.  We have changed the word “gender” to “sex in the title and in the lines that you kindly suggested.

Comment 2

  1. Figure 2 and fig 3 graphs and table formatting is not consistent. The grid format is present in fig 2 but not fig 3. 

Response 2

We adjusted figures 2 and 3 formatting to be the same (both with grids) as you suggested. They are now renamed  to figure 1 and 2 as we moved figure 1 to the supplementary material.We revised the tables formatting. We appreciate the comments.

Reviewer 4 Report

Comments and Suggestions for Authors

The study is well designed. The manuscript is well structured, and the ideas are well presented. However, if some aspects were clarified, the quality of the article could be improved.

In section 2.2.1. the authors could say (if only for information) how many changes the reconfiguration of stages implied (in figures or in percentages).

Figure 1 is of poor quality, could they provide another one with higher resolution and a better font size? Figure legends 2 and 3 are also of poor quality.

Regarding the types of treatments performed in each group, somewhere in the methodology or discussion it would be useful to clarify certain aspects (or cite as a weakness of the study):

i) Are they the same from start to finish (*) in each patient within each case series? What if we compare both countries?

(*) Same order of administration of surgery, radiotherapy or chemotherapy. Same dosimetry in radiotherapy treatment, same chemotherapy agents, etc. Has it been changing over the years, since the cases started to be registered?

ii) Are the health-systems and public health programs in UK and USA comparable? For example, are there screening programs in both countries for breast cancer (by systematic mammography), or for colorectal carcinoma (by fecal blood screening), or for relatives when there are specific cases of cancer in the family?

With regard to the demographic characteristics of the patient registry, have the migratory movements of recent years been taken into account, and is there any influence over the years?

It would also be interesting if it can be explicitly stated whether the diagnostic procedures and algorithms are the same, or whether they have changed over the years (or whether they are different in the UK and USA): selected imaging tests and order of use, confirmation by immunohistochemistry in tissue samples for pathological diagnosis, etc.

Author Response

Comment 1

The study is well designed. The manuscript is well structured, and the ideas are well presented. However, if some aspects were clarified, the quality of the article could be improved.

In section 2.2.1. the authors could say (if only for information) how many changes the reconfiguration of stages implied (in figures or in percentages).

Response 1

Thank you for taking the time to review our manuscript. We appreciate agreeing with the manuscript’s design, structure, and ideas. Regarding reconfiguring stages of NCRAS to align with SEER staging, the original stages were as follows: Stage I (34%), Stage II (13.5%), Stage III (18%) and Stage IV (34.5%) Then it was changed by combining stage I and II to be localized (47.5%), stage III (18%) to be regional and stage IV to be distant (34.5%). We included these numbers in section 2.2.1. as suggested.

Comment 2

Figure 1 is of poor quality, could they provide another one with higher resolution and a better font size? Figure legends 2 and 3 are also of poor quality.

Response 2

We improved the quality of figures 1,2, and 3 as you kindly suggested. We moved figure 1 to the supplementary section as well. We appreciate the comments.

Comment 3

Regarding the types of treatments performed in each group, somewhere in the methodology or discussion it would be useful to clarify certain aspects (or cite as a weakness of the study):

  1. i) Are they the same from start to finish (*) in each patient within each case series? What if we compare both countries?

(*) Same order of administration of surgery, radiotherapy or chemotherapy. Same dosimetry in radiotherapy treatment, same chemotherapy agents, etc. Has it been changing over the years, since the cases started to be registered?

Response 3

We agree that treatments (e.g., surgery, radiotherapy and chemotherapy) and their sequence are important variables and valuable information. However, The UK NCRAS database lacked these variables. The U.S. SEER database had treatment variables only for surgery types, administration of chemotherapy (Yes/No) with no details to specific classes, radiotherapy types as well as the sequence of treatment. However, these treatment variables in SEER have some inaccuracies and missing variables. SEER does not recommend using these variables due to these limitations as mentioned on the SEER website (SEER Treatment Data Limitations (November 2022 Submission) - SEER Data & Software (cancer.gov) We already cited this as a limitation in our manuscript (DOI:10.1097/MLR.0000000000000073). However, we don't believe there is a systematic bias between males and females' treatment of NEN in the USA and UK

Comment 4

  1. ii) Are the health-systems and public health programs in UK and USA comparable? For example, are there screening programs in both countries for breast cancer (by systematic mammography), or for colorectal carcinoma (by fecal blood screening), or for relatives when there are specific cases of cancer in the family?

With regard to the demographic characteristics of the patient registry, have the migratory movements of recent years been taken into account, and is there any influence over the years?

Response 4

Neither UK NCRAS nor SEER Databases have any data on migration, and it will be difficult to account for that. We have mentioned these limitations in the limitations section of the manuscript.

Healthcare systems in the UK and USA are not the same but there is unlikely as systematic bias in treatment of males and females. Screening programs for cancers such as colorectal carcinoma exist in both countries and are similar between UK and USA.

Comment 5

It would also be interesting if it can be explicitly stated whether the diagnostic procedures and algorithms are the same, or whether they have changed over the years (or whether they are different in the UK and USA): selected imaging tests and order of use, confirmation by immunohistochemistry in tissue samples for pathological diagnosis, etc.

Response 5

Both the USA and UK follow the WHO criteria for diagnosis of neuroendocrine neoplasms. We appreciate the time and effort spent reviewing the manuscript to improve it through these comments.

Round 2

Reviewer 1 Report

Comments and Suggestions for Authors

I have read the paper and the comments of all the reviewers with the changes implemented by our fellow researchers both in terms of the text and the iconography. The result is an excellent piece of writing that must be made available for reading in a prestigious journal like Cancers for an audience that wants to maintain a high medical-surgical cultural level. Excellent iconography, English and bibliography